# Can High-Fidelity Patient Simulation Be Used for Skill Development in Junior Undergraduate Students: A Quasi-Experimental Study

**DOI:** 10.3390/healthcare11152221

**Published:** 2023-08-07

**Authors:** Florence M. F. Wong, Alice M. L. Chan, Natalie P. M. Lee, Kevin K. H. Luk

**Affiliations:** 1School of Nursing, Tung Wah College, Hong Kong, China; alicechan@twc.edu.hk (A.M.L.C.); kevinluk@twc.edu.hk (K.K.H.L.); 2School of Nursing and Health Studies, Hong Kong Metropolitan University, Hong Kong, China; npmlee@hkmu.edu.hk

**Keywords:** high-fidelity patient simulation, problem-solving, clinical reasoning, healthcare training, nursing education

## Abstract

High-fidelity patient simulation (HFPS) is widely used in professional training to enhance students’ competence in clinical management. A guideline for HFPS provides a systematic approach to direct students to learning during the simulation process. Problem-solving (PS) and clinical reasoning (CR) skills are essential to developing students’ professional competence in safe and effective care. These two skills should be initiated in the early training. A structured guideline was developed for HFPS. This study aimed to investigate the effects of the structured HFPS guideline on the development of PS and CR skills in junior nursing students. The students were required to go through four sessions, pre-briefing, simulation design, facilitation, and debriefing, for the HFPS; the study utilized the Problem-Solving Inventory (PSI) and the Nurses’ Clinical Reasoning Scale (NCRS) to measure PS and CR abilities before and after HFPS. Bivariate analysis, a one-sample *t*-test, and an independent *t*-test were performed to evaluate the performance of the PS and CR skills during the two study periods. A total of 189 students were recruited, with 92 in the intervention group and 97 in the control group. The research assistant was responsible for student recruitment through email invitations and allocating the students into the control group or the intervention group. A Wilcoxon analysis was performed and revealed significant differences in PS and CR between the two groups (*p* < 0.001). The analytic results showed that the PSI, particularly in domains of Problem-Solving Confidence (PSC) (*p* < 0.001) and overall PS (*p* < 0.001), and the CR (*p* < 0.001) had significant improvement after HFPS, particularly in the intervention group. The study concluded that the structured HFPS guideline significantly improved the students’ problem-solving and clinical reasoning abilities. Nurse educators play an important role in providing explicit learning instructions in a simulation guideline that directs and guides students to learn at each stage of HFPS. The students can be directed to be engaged in their learning through HFPS to enhance their competence in knowledge and skill development (PS and CR) for their personal and professional development.

## 1. Introduction

In today’s healthcare settings, technology has advanced and become more complex. As a result, healthcare services emphasize evidence-based practice [1,2,3], and nurses are facing new challenges of immediate clinical management to provide a safer and higher quality of patient care [4,5]. Nurses are expected to exhibit more autonomy and proficiency in managing a wide array of intricate situations. Commencing early training to cultivate problem-solving (PS) and clinical reasoning (CR) skills is imperative to familiarize students with decision-making and clinical management in a clinical setting. Therefore, it is essential to train nursing students to attain the required competence standard for complex clinical situations, which includes problem-solving (PS) and clinical reasoning (CR) skills [5,6,7]. High-fidelity patient simulation (HFPS) is an innovative and efficacious pedagogical strategy that furnishes students with an immersive and simulated learning milieu, allowing them to apply their integrated knowledge and psychomotor skills to a realistic case scenario [8,9,10]. This authentic learning opportunity facilitates the development of higher-order intellectual skills, specifically PS and CR, which are crucial for more appropriate clinical decision-making [11]. An indispensable HFPS guideline is essential to concise directives that systematically guide students from preparation to debriefing, thereby promoting more effective learning [12]. The students must actively participate in the simulation, demonstrating their acquired knowledge and skills [8,10,13]. In the current nursing curriculum, HFPS is employed in various nursing disciplinary courses and clinical learning workshops to enhance students’ understanding of patients’ conditions and related treatment and care. An HFPS guideline provides systematic approaches to allow students to engage in their learning tasks, perform what they have learned, and evaluate how they have performed throughout the learning process of HFPS. It is necessary to examine the effects of structured HFPS guidelines on PS and CR abilities.

With the increased complexity of healthcare services, nurses are expected to have greater accountability and independence to provide appropriate quality patient care [2]. In current nursing education, students are required to equip themselves with sophisticated skills and knowledge to achieve the highest competency standards for better quality and safer patient care [14,15]. It is crucial to allow students to develop abilities in clinical judgement and decision-making by applying evidence-based knowledge and skills [16]. The cognitive processes of PS and CR are crucial for students to make informed decisions. However, classroom- and laboratory-based learning is limited in providing opportunities for students to apply their knowledge and skills in a practical context. Consequently, students may lack personal experience in performing these cognitive processes explicitly when confronted with real-world scenarios. This cognitive engagement necessitates the utilization of essential cognitive abilities, such as PS and CR, which enable students to make sound decisions considering multiple factors [16,17].

PS is the intellectual and analytic process that finds solutions to problems in specific contexts [18], while CR is the ability to integrate and apply the learned knowledge and experience, use and weigh the relevant evidence, and think critically about arguments until the final decision is made [19]. PS and CR are intertwined and crucial for competent practice and clinical judgements in nursing practice, embedding a series of critical thinking, integration of knowledge and skills, professional and personal experiences, and analysis [20,21]. Facing more responsibilities in clinical judgement and decision-making, nursing students are trained to manage complex situations using various teaching–learning modes.

HFPS is an advanced and innovative teaching–learning method that uses a computerised manikin in a simulated real-life scenario to allow students to integrate their knowledge and skills in their clinical decisions [22]. HFPS has been extensively and favourably used in educational and clinical training in recent decades to strengthen students’ PS and CR abilities [2,9]. This method provides a better learning environment for nursing students to practice in simulated diverse clinical situations [23]. Through HFPS, students practise self-directed learning in the preparation and processing of CR and PS for effective and appropriate decision-making, targeting complex and uncertain situations [20,21]. The students also experience their roles and responsibilities in a simulated clinical setting and understand more about their strengths and weaknesses, so that they can improve accordingly [23]. In that sense, CR and PS are vital abilities to enhance students’ competence in clinical performance [19,21]. Moreover, HFPS provides a learning environment for team collaboration through small group work. Students working in small groups can develop collaborative attributes, better learning motivation, and higher intellectual skills [24,25,26,27]. Therefore, students can interact and collaborate with their peers to exchange their learning experiences, enhancing their competence in nursing practice and teamwork skills in HFPS. Student learning can also be more dynamic and definite to incorporate more CR and PS. However, some previous studies have reported inconclusive results regarding the impact of HFPS on PS, most probably due to the absence of a structured educational approach [2,17]. Hence, it is crucial to provide nursing students with a comprehensive guideline that offers clear instruction and guidance to students for HFPS. As a result, an HFPS guideline incorporating education and management strategies was designed to enhance students’ knowledge acquisition and skill development. This study aimed to examine the effects of a structured HFPS guideline on PS and CR skills among undergraduate nursing students.

## 2. Materials and Methods

### 2.1. Design

This quasi-experimental study was conducted at a professional training institution.

### 2.2. Participants

Students who (1) were in their first or second year of an undergraduate programme and (2) were aged ≥18 years were recruited. Those who (1) had been enrolled in another course with HFPS or (2) had a previous clinical placement were excluded to avoid contamination. Students in the 3-year and 5-year Programmes of the Bachelor of Health Science (Honours) in Nursing (BHSN) and the Higher Diploma in Nursing (HDN) were invited. The BHSN programmes were designed to train registered nurses, while the HDN programme caters to enrolled nurses.

The sample size was calculated to reach a desired power of 0.95 and a type I error of 0.05, with an effect size of 0.5 using G*Power software (version 3.1.9.7). The required calculated minimum number of participants was 176 students. The eligible students were requested to join a group at their preferred timeslot. When the group size reached 8 to 10 students, the research assistant (RA) would inform the educator. The RA was not involved in the implementation of HFPS.

### 2.3. A Structured HFPS Guideline as the Study Framework

The present study utilized a structured HFPS guideline, which was developed based on the Healthcare Simulation Standards of Best Practice (HSSOBP) by the International Nursing Association for Clinical Simulation and Learning [INACSL], drawing on clinical and academic expertise in HFPS [28]. The guideline was implemented to provide a systematic approach to guide students through simulated activities and consisted of four major sessions of HFPS, each with four standards: pre-briefing, simulation design, facilitation, and debriefing. Appendix A provides a detailed breakdown of the guideline employed in the control and intervention groups. Specifically, the students in the intervention group received the structured HFPS guideline, while those in the control group received the standard treatment with basic instructions for HFPS. During the pre-briefing session, the students were instructed to review the rules and regulations, learning objectives, and learning materials related to the simulated patient’s health problem and its associated medical and nursing care. Additionally, the students were familiarized with the simulation environment and equipment before proceeding to the role-playing session. The scenario was designed to facilitate the HFPS, during which the students were divided into three small groups and worked collaboratively to address the simulated patient’s health needs. In the debriefing session, the students were encouraged to reflect on their learning, based on their role-playing performance. To avoid contamination, two educators were assigned, one for the intervention group, and the other for the control group. The educators served as facilitators during the HFPS.

### 2.4. Instruments

#### 2.4.1. Problem-Solving Inventory (PSI)

The PSI developed by Heppner and Petersen [29] is used to measure individuals’ perceptions regarding their PS abilities and styles in daily life. It consists of 32 items scored on a 6-point Likert scale, ranging from 1 (strongly agree) to 6 (strongly disagree). The PSI includes three subscales: Problem-Solving Confidence (PSC) (11 items), Approach-Avoidance Style (AAS) (16 items), and Personal Control (PC) (5 items). The PSC subscale assesses self-perceived confidence, belief, and self-assurance in effectively solving problems. Higher scores indicate lower levels of PS confidence. The AAS subscale measures an individual’s response tendency to approach or avoid problems. Higher scores reflect avoidance rather than approaching problems. The PC subscale assesses elements of self-control of emotions and behaviours. Higher scores indicate a more negative perception of personal control of problems. The total PSI score ranges from 32 to 192. Lower total PSI scores indicate more functional PS abilities. The reliability of the subscales and the overall scale was good to very good, with raw coefficient alphas of PSC, AAS, PC, and overall PS at 0.819, 0.810, 0.710, and 0.892, respectively.

#### 2.4.2. Nurses Clinical Reasoning Scale (NCRS)

The NCRS developed by Liou et al. [30] assesses students’ CR competence. This self-reported tool includes 15 items scored on a 5-point Likert scale ranging from 1 (strongly disagree) to 5 (strongly agree). Higher scores indicate higher CR competence. The Cronbach’s alpha of the NCRS was 0.952, indicating a high level of reliability.

#### 2.4.3. Data Collection

The recruitment of participants was conducted by the research assistant (RA) through email correspondence, wherein the purpose and details of the study were conveyed to prospective students. The RA was also responsible for the allocation of eligible participants into either the control or intervention group. The intervention group was instructed to utilize the structured HFPS guideline, while the control group followed the standard HFPS instruction. The HFPS was facilitated by two experienced researchers with more than five years of experience in teaching HFPS. To avoid contamination, the researchers acted as facilitators for the intervention and control groups, respectively, in separate venues. During the HFPS, the students were divided into three small groups, each with a 20 min role-playing session to provide care for the simulated patient. While one group performed in the simulation, the other two groups observed and provided feedback. Subsequently, debriefing was conducted for the purpose of performance improvement. The flow of data collection and study implementation is presented in Figure 1.

#### 2.4.4. Ethical Considerations

Ethical approval (REC2021102) was sought from the Institutional Research Committee. Informed consent was obtained from the students after the study purpose and procedure were explained to them. All the data related to personal information were kept confidential.

#### 2.4.5. Data Analysis

Statistical analyses were performed using IBM SPSS version 26. Descriptive statistics were used for the demographic variables, including age, sex, study year, and study programme, and the outcomes of the intervention and control groups were presented separately. A paired *t*-test was performed to compare the PS and CR abilities before and after the HFPS. All the statistical tests involved were two-sided, and *p*-values of <0.05 were considered statistically significant.

## 3. Results

### 3.1. Student Characteristics

A total of 189 students participated in this study without attrition. The mean age was 20.56 years (standard deviation = 3.14). Table 1 shows the detailed demographic characteristics.

### 3.2. PS and CR Abilities of the Two Groups

There was a significant relationship between PS and CR abilities (before intervention: γ = −0.17, *p* = 0.020; after intervention: γ = −0.50, *p* < 0.001). This indicated that PS and CR had a significant negative relationship. Table 2 illustrates the descriptive results of the PS and CR abilities before and after the HFPS. The descriptive analysis showed that the PS and CR abilities had certain improvements after the HFPS. The bivariate analysis revealed that only age was negatively associated with the PSC subscale score (γ = −0.16, *p* = 0.027) and overall PSI score (γ = −0.147, *p* = 0.044).

### 3.3. Comparison between Two Periods (Table 3)

In the paired samples *t*-test (Table 3), there were significant differences observed in the PSC subscale score (*p* < 0.001), overall PSI score (*p* < 0.001), and NCRS score (*p* < 0.001) between the two periods of HFPS. In the two periods of HFPS, the PSI score and NCRS score more significantly changed in the intervention group [PSC subscale score (*p* < 0.001), overall PSI score (*p* = 0.011), and NCRS score (*p* < 0.001)] than in the control group [PSC subscale score (*p* < 0.001), overall PSI score (*p* = 0.011), and NCRS score (*p* = 0.014)].

The paired *t*-test revealed that the effect sizes, as measured by Cohen’s d, were found to be less than 0.2 in both AAS and PC, suggesting a very small effect. Conversely, in the other PS subscales and NCRS, the effect sizes ranged from 0.3 to 0.7, indicating a small to medium effect.

**Table 3 healthcare-11-02221-t003:** Comparisons of problem-solving and clinical reasoning abilities between two periods of high-fidelity patient simulation.

Paired Samples *t*-Test				
Periods	t	*p*	95% CI	Cohen d
**AAS**				
Control	1.55	0.125	−0.40 to 3.23	0.19
Intervention	0.13	0.899	−1.43 to 1.62	0.02
Overall	1.29	0.199	−0.41 to 1.96	0.11
**PSC**				
Control	6.01	<0.001 ***	2.20 to 4.37	0.58
Intervention	5.60	<0.001 ***	1.93 to 4.05	0.54
Overall	8.23	<0.001 ***	2.39 to 3.90	0.56
**PC**				
Control	1.70	0.093	−0.12 to 1.50	0.20
Intervention	0.29	0.776	−0.52 to 0.69	0.03
Overall	1.55	0.123	−0.11 to 0.90	0.12
**Overall PS**				
Control	3.40	0.001 **	2.24 to 8.54	0.40
Intervention	2.59	0.011 **	0.74 to 5.61	0.26
Overall	4.27	<0.001 ***	2.32 to 6.30	0.34
**NCRS**				
Control	−2.51	0.014 *	−5.15 to −0.61	0.30
Intervention	−5.83	<0.001 ***	−7.75 to −3.81	0.69
Overall	−5.61	<0.001 ***	−5.80 to −2.78	0.47

AAS: approach–avoidance style, PSC: problem-solving confidence, PC: personal control, Overall PS: problem-solving ability, NCRS: Nurses’ Clinical Reasoning Scale. * *p* < 0.05; ** *p* < 0.01, *** *p* < 0.001.

### 3.4. Comparison between Two Groups (Table 4)

A Wilcoxon analysis was performed, due to the absence of a normal distribution. The findings indicate statistically significant variations in PS and CR measures between the two groups, as demonstrated by a z-score of −5.385 with a *p*-value of less than 0.001 for PS, and a z-score of −5.92 with a *p*-value of less than 0.001 for CR.

In the independent sample *t*-test (Table 4), there were significant differences found in the PSI scores, including the AAS subscale score (*p* = 0.016), PSC subscale score (*p* = 0.034), and overall PSI score (*p* = 0.012). Meanwhile, the NCRS score before the HFPS did not significantly differ between the two groups. However, there was no significant difference in all the PSI subscale scores and the NCRS scores between the two groups after the HFPS.

The independence *t*-test revealed that the effect sizes, as measured by Cohen’s d, were found to be less than 0.4 in all the subscales of PS and NCRS, indicating a very small to small effect.

**Table 4 healthcare-11-02221-t004:** Comparison of problem-solving ability and clinical reasoning between control and intervention groups.

	Independent *t*-Test			
Periods	t	*p*	95% CI	Cohen d
**AAS**				
Pre-simulation	2.42	0.016 *	0.37 to 3.57	0.35
Post-simulation	0.55	0.583	−1.70 to 3.01	0.08
**PSC**				
Pre-simulation	2.14	0.034 *	0.12 to 2.93	0.31
Post-simulation	1.35	0.179	−0.57 to 3.02	0.20
**PC**				
Pre-simulation	0.71	0.481	−0.58 to 1.22	0.10
Post-simulation	−0.57	0.571	−1.26 to 0.70	0.08
**Overall PS**				
Pre-simulation	2.54	0.012 *	0.85 to 6.79	0.37
Post-simulation	0.74	0.461	−2.67 to 5.88	0.11
**NCRS**				
Pre-simulation	0.25	0.800	−3.28 to −0.08	0.04
Post-simulation	−1.85	0.066	−1.34 to 1.81	0.27

AAS: approach–avoidance style, PSC: problem-solving confidence, PC: personal control, Overall PS: problem-solving ability, NCRS: Nurses’ Clinical Reasoning Scale. * *p* < 0.05.

## 4. Discussion

This study revealed that the structured HFPS guideline effectively enhances the PS and CR abilities of junior students. Comparing the PS domains and CR between the two periods, the PSC subscale score (*p* < 0.001), overall PSI score (*p* < 0.001), and NCRS score (*p* < 0.001) significantly improved. Notably, the HFPS significantly improves students’ PS confidence, leading to improved PS ability and increased CR ability. The HFPS provides a good learning and practice environment for students to apply their knowledge and skills to the simulated patient [9,12,23]. Students’ PS and CR improvement is attributed to the guideline with four sessions—pre-briefing, simulation design, facilitation, and debriefing. The structured guideline effectively provides adequate instruction to facilitate student learning and guidance to allow students to be engaged in each HFPS session. Although HFPS improves PS and CR abilities on its own, following the HFPS guidelines can further enhance students’ PS and CR development. It is crucial for students to actively participate in their learning and remain fully engaged in the learning environment’s activities of the entire HFPS. In the pre-briefing, the students receive related learning materials to prepare, and an orientation to the simulation environment for their role-play in the HFPS. Through the students’ self-study and environmental recognition, they were able to increase their knowledge and apply their skills, so they had more confidence to handle clinical situations. Pre-reading relevant learning materials related to the simulated patient can foster students’ confidence in PS to cope with problems more effectively, and hence develop their clinical judgement ability [7,9]. Therefore, the learning materials, which should be interesting and simple, enable students to understand their roles and responsibilities, comprehend the case scenario for HFPS, and grasp relevant evidence-based information. This will better prepare them for the HFPS. Prior to the implementation of the HFPS, significant differences were observed between the two groups in terms of AAS subscale scores (*p* = 0.016), PSC subscale scores (*p* = 0.034), and overall PSI scores (*p* = 0.012). The control group exhibited more avoidance behaviours and lower confidence levels in PS compared to the intervention group. It is worth noting that pre-reading and orientation activities can enhance students’ confidence in PS, enabling them to effectively cope with problems and develop their clinical judgment abilities [7,9]. Thus, adequate preparation not only increases students’ awareness and application of acquired knowledge and skills but also facilitates the expression of high levels of critical thinking and PS abilities, resulting in the provision of safe and appropriate patient care during HFPS [4,20]. Further, the structured guideline appears to positively affect the students’ CR abilities.

Substantial evidence shows that HFPS is a crucial component in enhancing clinical experience and clinical management skills [16,31]. A case scenario designed with a specific health problem for the HFPS allows students to experience clinical practice and patient care. During the role-playing session, students are required to be more engaged in the simulated clinical scenario. Adequate preparation enhances students’ awareness and application of learned knowledge and skills in the simulated clinical situation. It also helps them express high levels of critical thinking and PS abilities for the provision of safe and appropriate patient care during HFPS [32]. The simulated patient’s concerns are challenging, and their responses are promising, increasing their confidence in real clinical practice. Accordingly, the PS and CR abilities of the two groups in this study improved. HFPS also focuses on teamwork and enables students to provide immediate management according to the client’s needs and conditions [9]. Working in a small group is beneficial in enhancing knowledge and developing skills, including high intellectual skills, such as communication, PS, critical thinking, and collaborative skills [24,25,26]. Since students are required to work as a team in HFPS, they collaborate with other team members for decision-making and develop their personal and professional strengths together.

During the role-playing session and the debriefing, the facilitator plays an important role in engaging students in learning through the HFPS more effectively [23,31]. The facilitator provides timely guidance for students to perform and react to problems in various environmental diversions. In the debriefing, the facilitator allows students to reflect on their learning and provides appropriate guidance for more effective clinical judgments and decision-making accordingly [13]. Both self and peer reflection can further enhance students’ understanding to perform more competently.

The structured HFPS guideline with adequate instruction, learning materials, and expected learning outcomes is essential to positively provide students with systematic direction and support so that the students achieve PS and CR development more effectively. This study successfully demonstrated the benefits of the structured HFPS guideline for student learning and skill development, including better preparation to increase knowledge acquisition and skill application, development of collaborative attributes, and enhancement of higher intellectual skills, such as PS and CR. Therefore, the structured guideline should be added to the courses with HFPS in the nursing curriculum. The results also promote the awareness of nurse educators to design HFPS guidelines and further enhance students’ PS and CR abilities, which are crucial for clinical judgement and decision-making, even in junior years. Therefore, early training cultivates PS and CR abilities in nursing students, enabling them to become familiar with decision-making and clinical management in the healthcare setting. It will produce a significant impact on the future clinical practice and nursing profession of the students.

The structured guideline is useful to assist students in achieving their learning outcomes in a systematic manner. It is important to periodically review the content of the guideline to ensure that it meets the specific learning needs of students at different levels of education. Additionally, conducting studies to evaluate the effectiveness of HFPS in enhancing clinical practice would be highly beneficial in gaining a deeper understanding of students’ learning requirements and providing them with additional support to manage complex clinical situations.

### Strengths and Limitations

This study enables a direct comparison of the outcome measures between two stages of HFPS, providing reliable and accurate evidence of the intervention’s effects. Two educators with adequate experience in HFPS provide timely facilitation during HFPS. However, comprehending the development of students’ PS and CR necessitates a greater understanding of the sustainable impact of HFPS. In addition, the recruitment of students at a single professional training institution limits the generalisability of the results and the ability to draw causal inferences. To increase generalisability, a similar study should be conducted in multiple centres. The effectiveness of the intervention may be limited due to the lack of randomization, which can also introduce selection bias by creating systematic differences between the groups being compared.

The structured guideline was developed to guide students to achieve their learning outcomes in a systematic way. The content of the guideline should be reviewed to meet the specific learning needs of students at various years. Studies to examine the application of HFPS beneficial to clinical practice are recommended to further understand students’ learning needs and provide more support for them to handle more complex clinical situations.

## 5. Conclusions

The structured HFPS guideline in this study provides clear directions for students to prepare and enhance their knowledge and skills in clinical judgement and decision-making. Additionally, the HFPS guideline significantly improves PS and CR abilities before and after participating in HFPS. Throughout the learning process of HFPS, students not only apply their knowledge and skills but also increase their awareness and application of appropriate practice through a series of PS and CR. This study has significant implications for nursing education. As the use of HFPS continues to increase in professional training, including nursing, the findings highlight the importance of an HFPS guideline to improve student learning and competence in practising through HFPS, leading to better development of PS and CR abilities. Therefore, an HFPS guideline is crucial in enhancing students’ personal and professional development for more appropriate and safer clinical decisions and judgements.

## Figures and Tables

**Figure 1 healthcare-11-02221-f001:**
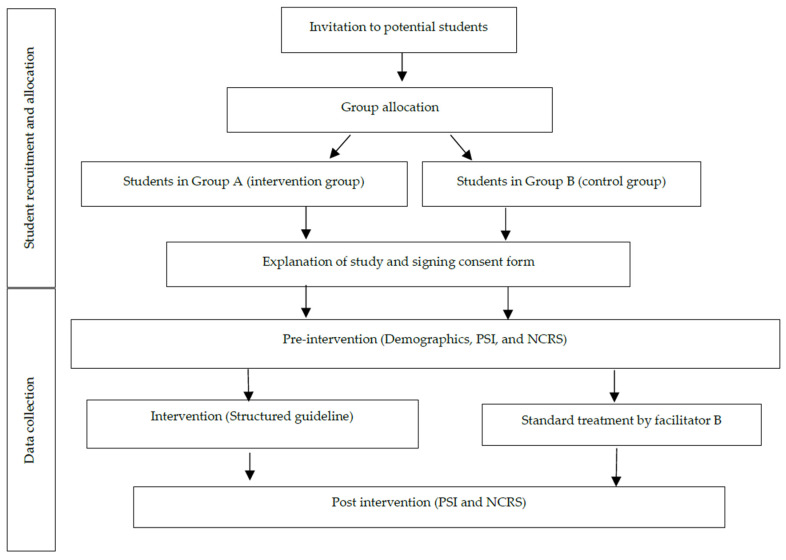
Flow of data collection and study implementation.

**Table 1 healthcare-11-02221-t001:** Student demographic characteristics.

	Overall (n = 189)		Intervention (n = 92)		Control (n = 97)		*p*
	n	%	n	%	n	%	
Gender							0.211
Male	51	27	21	22.8	30	30.9	
Female	138	73	71	77.2	67	69.1	
Age							0.027 *
18–20	135	71.4	73	79.3	62	63.9	
21–24	35	18.5	12	13.0	23	23.7	
25–27	11	5.8	4	4.3	7	7.2	
28–30	5	2.6	3	3.3	2	2.1	
>30	3	1.6	73	79.3	3	3.1	
Programme							0.023 *
HDN	55	29.1	19	20.7	36	37.1	
BHSN (5-year programme)	118	62.4	64	69.6	54	55.7	
BHSN (3-year programme)	16	8.5	9	9.8	7	7.2	
Study Year							0.919
1	102	54.0	50	54.3	52	53.6	
2	87	46.0	42	45.7	45	46.4	

HDN: Higher Diploma in Nursing, BHSN: Bachelor of Health Science (Honours) in Nursing, * *p* < 0.05.

**Table 2 healthcare-11-02221-t002:** Descriptive results.

**Overall (n = 189)**							
	**Pre**		**After**		**Changes**		** *p* **
	**Mean**	**SD**	**Mean**	**SD**	**Mean**	**SD**	
PS ability							
AAS	50.78	5.07	50.68	7.82	−0.77	8.25	0.199
PSC	31.98	5.02	28.99	6.07	−3.14	5.25	<0.001 ***
PC	16.34	2.98	16.25	3.2	−0.4	3.53	1.55
Overall PS	99.1	9.93	95.92	13.87	−4.31	13.89	<0.001 ***
NCRS	48.34	7.89	54.12	8.85	4.29	10.52	<0.001 ***
**Intervention Group (n = 92)**							
	**Pre**		**After**		**Changes**		** *p* **
	**Mean**	**SD**	**Mean**	**SD**	**Mean**	**SD**	
PS ability							
AAS	50.78	5.07	50.68	7.82	−0.1	7.36	0.899
PSC	31.98	5.02	28.99	6.07	−2.99	5.12	<0.001 ***
PC	16.34	2.98	16.25	3.2	−0.09	2.93	0.29
Overall PS	99.1	9.93	95.92	13.87	−3.17	11.77	0.011 **
NCRS	48.34	7.89	54.12	8.85	5.78	9.51	<0.001 ***
**Control Group (n = 97)**							
	**Pre**		**After**		**Changes**		** *p* **
	**Mean**	**SD**	**Mean**	**SD**	**Mean**	**SD**	
PS ability							
AAS	52.75	6.04	51.34	8.53	−1.41	9	0.125
PSC	33.51	4.79	30.22	6.43	−3.29	5.39	<0.001 ***
PC	16.66	3.28	15.97	3.59	−0.69	4.01	1.7
Overall PS	102.92	10.69	97.53	15.79	−5.39	15.62	0.001 **
NCRS	48.65	9.01	51.53	10.36	2.88	11.27	0.014 *

AAS: approach–avoidance style, PSC: problem-solving confidence, PC: personal control, Overall PS: problem-solving ability, NCRS: Nurses’ Clinical Reasoning Scale. * *p* < 0.05; ** *p* < 0.01, *** *p* < 0.001.

## Data Availability

The data presented in this study are available on request from the corresponding author. The data are not publicly available due to keep the confidentiality.

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
