# Peer review of "Can High-Fidelity Patient Simulation Be Used for Skill Development in Junior Undergraduate Students: A Quasi-Experimental Study"

_healthcare, 2023, doi:10.3390/healthcare11152221_

Round 1
Reviewer 1 Report
This is an interesting paper - there are considerable strengths such as the evidence upon which to base teaching and learning approaches and adapting to changing environments (ie lack of placements, blocks to certain clinical areas). I add these observations to support and add improvements to develop this paper.
1. The context revolves around the use f HFPS and has a few conflicting comments. Further the contexts of the nurses might be useful - the mean age is very young and they are year 1 so it is not clear how much if any clinical practice they had and how this may have impacted their ability to clinical reason. A description of the scenario and type of reasoning expected would add clarity (linked to line 38 & 39 - complex situations)
2. Small Typos - 'e' in line 45
3. Line 78 HFPS used successfully to improve PS & CR, yet line 91 PS unaffected by HFPS in nursing students - perhaps needs to be clearer on the earlier comment as to who you are referring.
3. What limitations were there of future work ie look to year 3 students and more complex clinical situations?
Author Response
Thank you sincerely for your invaluable feedback, which has greatly contributed to the improvement of the manuscript.
This is an interesting paper - there are considerable strengths such as the evidence upon which to base teaching and learning approaches and adapting to changing environments (ie lack of placements, blocks to certain clinical areas). I add these observations to support and add improvements to develop this paper.
- The context revolves around the use f HFPS and has a few conflicting comments. Further the contexts of the nurses might be useful - the mean age is very young and they are year 1 so it is not clear how much if any clinical practice they had and how this may have impacted their ability to clinical reason. A description of the scenario and type of reasoning expected would add clarity (linked to line 38 & 39 - complex situations)
Response: Student participants who had had no clinical experience were recruited in the study. The HFPS is usually employed in the senior year students. This study tried to evaluate how the junior students could be benefited through a structured HFPS guideline. Early training to develop PS and CR abilities should be started so that students are able to make familiar with decision making and clinical management in the clinical environment. The selection criteria of participants have been revised. The study results about the effects on PS and CR abilities will benefit students to their future clinical practice and nursing career. Such explanation has been added in the “Discussion” part. Please refer to p.8. for the revision highlighted in yellow.
- Small Typos - 'e' in line 45
Response: removed.
- Line 78 HFPS used successfully to improve PS & CR, yet line 91 PS unaffected by HFPS in nursing students - perhaps needs to be clearer on the earlier comment as to who you are referring.
Response: The lines starting from 91 have been rewritten to clearly describe the previous studies reported inconclusive results most probably due to the absence of structured or clear guideline for HFPS. Please refer to p.3 for the revision highted in yellow.
- What limitations were there of future work ie look to year 3 students and more complex clinical situations?
Response: Additional content to address your valuable comments in the Discussion. Please refer to page 8 highlighted in yellow for the information.
Reviewer 2 Report
I think it is important to use HP simulation to improve PS and CR for nursing students. I'd like to make a few comments.
1. There is a lack of consistency between the research title and the contents of the paper.
- Introduction does not provide a background description of the 'After Coronavirus Pandemic' mentioned in the study title.
- How does Coronavirus Pandemic affect the relationship between HFPS use and the improvement of Problem-solving and clinical reasoning skills in nursing students?
- In the title, the subjects are First-Year Undergraduates, but the actual subjects are inconsistent, including first and second graders.
2. Under Table 1, describe the full name for the abbreviations of HDN, BHSc.
3. Table 2 seems to have tested the difference between PS and CR before and after the HFPS, so complete the table by presenting the statistics or significance levels.
4. How many people were there in the Experimental(intervention) group and Control group? It appears that there was virtually no Control group in this study.
It is hard to believe that there is a difference simply by comparing the changes in PS and CR in the intervention group. Because just doing the same questionnaire before and after the HP simulation would have given them some information that would have affected the second questionnaire. Also, various educational information delivered during the simulation process also affected, so I think it is very dangerous to interpret it as the effect of simulation alone.
In conclusion, this research design belongs to the type of primitive experimental design.
It seems that the research design needs to be revised.
Thank you for your efforts.
Author Response
Thank you sincerely for your invaluable feedback, which has greatly contributed to the improvement of the manuscript.
I think it is important to use HP simulation to improve PS and CR for nursing students. I'd like to make a few comments.
- There is a lack of consistency between the research title and the contents of the paper.
- Introduction does not provide a background description of the 'After Coronavirus Pandemic' mentioned in the study title.
Response: The study title has been revised to align with the context. The revised title is “Can High-Fidelity Patient Simulation Be Used for Skill Development in Junior Undergraduate Students: A Quasi-Experimental Study.”
- How does Coronavirus Pandemic affect the relationship between HFPS use and the improvement of Problem-solving and clinical reasoning skills in nursing students?
Response: Agree with your comment. The title is revised as “Can High-Fidelity Patient Simulation Be Used for Skill Development in Junior Undergraduate Students: A Quasi-Experimental Study.”
- In the title, the subjects are First-Year Undergraduates, but the actual subjects are inconsistent, including first and second graders.
Response: The student participants from the first and second years were eligible to meet the recruitment criteria. The “First-Year” in the title and in the content has been revised to “Junior”.
- Under Table 1, describe the full name for the abbreviations of HDN, BHSc.
Response: The full names of HDN and BHSC have been added as advised.
- Table 2 seems to have tested the difference between PS and CR before and after the HFPS, so complete the table by presenting the statistics or significance levels.
Response: The statistical results between problem-solving and clinical reasoning were found to have significant negative relationship. The results have been added in the point #3.2. The results of significance levels have been also added in Table 2 highlighted in yellow. Please refer to page 5 for the revision.
- How many people were there in the Experimental(intervention) group and Control group? It appears that there was virtually no Control group in this study.
Response: There were 92 students in the intervention group and 97 in the control group. Information about the group allocation and study procedure were described more detail in the manuscript.
It is hard to believe that there is a difference simply by comparing the changes in PS and CR in the intervention group. Because just doing the same questionnaire before and after the HP simulation would have given them some information that would have affected the second questionnaire. Also, various educational information delivered during the simulation process also affected, so I think it is very dangerous to interpret it as the effect of simulation alone.
Response: Thank you for your reminder. Our team considered that this study should be a quasi-experimental study. Although there was no randomization, the structured guideline was the intervention to examine its effects on problem-solving and clinical reasoning. Therefore, the results were not only for the effect of simulation alone. The intervention group received structured guideline and the control group received standard instruction. The 2.4.3 Data collection section has been revised to describe the procedure more clearly. The Table S1 has been added to increase understanding of the two treatments.
In conclusion, this research design belongs to the type of primitive experimental design.
It seems that the research design needs to be revised.
Response: The research design has been revised to a qusai-experimental design because the study included 2 groups, one intervention and one control groups. The data were collected before and after HFPS.
Reviewer 3 Report
See attached

Quite good
Author Response
Thank you sincerely for your invaluable feedback, which has greatly contributed to the improvement of the manuscript. Please find the attachment for the responses to your comments.

Round 2
Reviewer 2 Report
No more comments.